

SciPost Phys. 1(1), 001 (2016)

# Quantum quenches to the attractive one-dimensional Bose gas: exact results

L. Piroli[1*], P. Calabrese[1], F.H.L. Essler[2]

**1** SISSA and INFN, via Bonomea 265, 34136 Trieste, Italy.
**2** The Rudolf Peierls Centre for Theoretical Physics, Oxford University, Oxford, OX1 3NP, United Kingdom.

* lpiroli@sissa.it

## Abstract

We study quantum quenches to the one-dimensional Bose gas with attractive interactions in the case when the initial state is an ideal one-dimensional Bose condensate. We focus on properties of the stationary state reached at late times after the quench. This displays a finite density of multi-particle bound states, whose rapidity distribution is determined exactly by means of the quench action method. We discuss the relevance of the multi-particle bound states for the physical properties of the system, computing in particular the stationary value of the local pair correlation function $g_2$.



# 1 Introduction

Strongly correlated many-body quantum systems are often outside the range of applicability of standard perturbative methods. While being at the root of many interesting and some-times surprising physical effects, this results in huge computational challenges, which are most prominent in the study of the non-equilibrium dynamics of many-body quantum systems. This active field of research has attracted increasing attention over the last decade, also due to the enormous experimental advances in cold atomic physics [1–3]. Indeed, highly isolated many-body quantum systems can now be realised in cold atomic laboratories, where the high experimental control allows to directly probe their unitary time evolution [4–13].

A simple paradigm to study the non-equilibrium dynamics of closed many-body quantum systems is that of a quantum quench [14]: a system is prepared in an initial state (usually the ground state of some Hamiltonian $H_0$) and it is subsequently time evolved with a local Hamiltonian $H$. In the past years, as a result of a huge theoretical effort (see the reviews [2, 15–21] and references therein), a robust picture has emerged: at long times after the quench, and in the thermodynamic limit, expectation values of *local* observables become stationary. For a generic system, these stationary values are those of a thermal Gibbs ensemble with the effective temperature being fixed by the energy density in the initial state [22].

A different behaviour is observed for integrable quantum systems, where an infinite set of local conserved charges constrains the non-equilibrium dynamics. In this case, long times after the quench, local properties of the systems are captured by a generalised Gibbs ensemble (GGE) [23], which is a natural extension of the Gibbs density matrix taking into account a complete set of local or quasi-local conserved charges.

The initial focus was on the role played by (ultra-)local conservation laws in integrable quantum spin chains [24–30], while more recent works have clarified the role by sets of novel, quasi-local charges [31–41]. It has been shown recently that they have to be taken into account in the GGE construction in order to obtain a correct description of local properties of the steady state [44,45]. Quasi-local conservation laws and their relevance for the GGE have also recently

been discussed in the framework of integrable quantum field theories [48,49]. These works have demonstrated that the problem of determining a complete set of local or quasi-local conserved charges is generally non-trivial.

A different approach to calculating expectation values of local correlators in the stationary state was introduced in Ref. [50]. It is the so called quench action method (QAM), a.k.a. representative eigenstate approach and it does not rely on the knowledge of the conserved charges of the system. Within this method, the local properties at large times are effectively described by a single eigenstate of the post-quench Hamiltonian. The QAM has now been applied to a variety of quantum quenches, from one dimensional Bose gases [51–56] to spin chains [57–61] and integrable quantum field theories [62,63], see Ref. [20] for a recent review.

One of the most interesting aspects of non-equilibrium dynamics in integrable systems is the possibility of realising non-thermal, stable states of matter by following the unitary time evolution after a quantum quench. Indeed, the steady state often exhibits properties that are qualitatively different from those of thermal states of the post-quench Hamiltonian. The QAM provides a powerful tool to theoretically investigate these properties in experimentally relevant settings.

In this paper we study the quantum quench from an ideal Bose condensate to the Lieb-Liniger model with arbitrary attractive interactions. A brief account of our results was previously given in Ref. [56]. The interest in this quench lies in its experimental feasibility as well as in the intriguing features of the stationary state, which features finite densities of multi-particle bound states. Our treatment, based on the quench action method, allows us to study their dependence on the final interaction strength and discuss their relevance for the physical properties of the system. In particular, as a meaningful example, we consider the local pair correlation function $g_2$, which we compute exactly.

The structure of the stationary state is very different from the super Tonks-Girardeau gas, which is obtained by quenching the one-dimensional Bose gas from infinitely repulsive to infinitely attractive interaction [64–70]. The super Tonks-Girardeau gas features no bound states, even though it is more strongly correlated than the infinitely repulsive Tonks-Girardeau gas, as has been observed experimentally [66]. As we argued in [56], the physical properties of the post-quench stationary state reached in our quench protocol could be probed in ultracold atoms experiments, and the multi-particle bound states observed by the presence of different"light-cones" in the spreading of local correlations following a local quantum quench [71].

In this work we present a detailed derivation of the results previously announced in Ref. [56]. The remainder of this manuscript is organised as follows. In section 2 we introduce the Lieb-Liniger model and the quench protocol that we consider. The quench action method is reviewed in section 3, and its application to our quench problem is detailed. In section 4 the equations describing the post-quench stationary state are derived. Their solution is then obtained in section 5, and a discussion of its properties is presented. In section 6 we address the calculation of expectation values of certain local operators on the post-quench stationary state, and we explicitly compute the local pair correlation function $g_2$. Finally, our conclusions are presented in section 7. For the sake of clarity, some technical aspects of our work are consigned to several appendices.

## 2  The Lieb-Liniger model

### 2.1  The Hamiltonian and the eigenstates

We consider the Lieb-Liniger model [72], consisting of $N$ interacting bosons on a one-dimensional ring of circumference $L$. The Hamiltonian reads

$$H_{LL}^N = -\frac{\hbar^2}{2m}\sum_{j=1}^N \frac{\partial^2}{\partial x_j^2} + 2c\sum_{j<k}\delta(x_j - x_k),\tag{1}$$

where $m$ is the mass of the bosons, and $c = -\hbar^2/ma_{1D}$ is the interaction strength. Here $a_{1D}$ is the 1D effective scattering length [73] which can be tuned via Feshbach resonances [74]. In the following we fix $\hbar = 2m = 1$. The second quantized form of the Hamiltonian is

$$H_{LL} = \int_0^L \mathrm{d}x\left\{\partial_x\Psi^\dagger(x)\partial_x\Psi(x) + c\Psi^\dagger(x)\Psi^\dagger(x)\Psi(x)\Psi(x)\right\},\tag{2}$$

where $\Psi^\dagger$, $\Psi$ are complex bosonic fields satisfying $[\Psi(x),\Psi^\dagger(y)] = \delta(x-y)$.

The Hamiltonian (1) can be exactly diagonalised for all values of $c$ using the Bethe ansatz method [42,72]. Throughout this work we will consider the attractive regime $c < 0$ and use notations $\bar{c} = -c > 0$. We furthermore define a dimensionless coupling constant by

$$\gamma = \frac{\bar{c}}{D},\quad D = \frac{N}{L}.\tag{3}$$

A general $N$-particle energy eigenstate is parametrized by a set of $N$ complex rapidities $\{\lambda_j\}_{j=1}^N$, satisfying the following system of Bethe equations

$$e^{-i\lambda_j L} = \prod_{\substack{k\neq j\\k=1}}^N \frac{\lambda_k - \lambda_j - i\bar{c}}{\lambda_k - \lambda_j + i\bar{c}},\quad j = 1,\ldots,N.\tag{4}$$

The wave function of the eigenstate corresponding to the set of rapidities $\{\lambda_j\}_{j=1}^N$ is then

$$\psi_N(x_1,\ldots,x_N|\{\lambda_j\}_{j=1}^N) = \frac{1}{\sqrt{N}}\sum_{P\in\mathscr{S}_N}e^{i\sum_j x_j\lambda_{P_j}}\prod_{j>k}\frac{\lambda_{P_j}-\lambda_{P_k}+i\bar{c}\,\mathrm{sgn}(x_j-x_k)}{\lambda_{P_j}-\lambda_{P_k}},\tag{5}$$

where the sum is over all the permutations of the rapidities. Eqns (4) can be rewritten in logarithmic form as

$$\lambda_j L - 2\sum_{k=1}^N \arctan\left(\frac{\lambda_j - \lambda_k}{\bar{c}}\right) = 2\pi I_j,\quad j = 1,\ldots,N,\tag{6}$$

where the quantum numbers $\{I_j\}_{j=1}^N$ are integer (half-odd integer) for $N$ odd (even).

In the attractive regime the solutions of (6) organize themselves into mutually disjoint patterns in the complex rapidity plane called "strings" [75,76]. For a given $N$ particle state, we indicate with $\mathscr{N}_s$ the total number of strings and with $N_j$ the number of $j$-strings, i.e. the strings containing $j$ particles ($1 \le j \le N$) so that

$$N = \sum_j jN_j,\qquad \mathscr{N}_s = \sum_j N_j.\tag{7}$$

The rapidities within a single $j$-string are parametrized as [87]

$$\lambda_\alpha^{j,a} = \lambda_\alpha^j + \frac{i\bar{c}}{2}(j+1-2a) + i\delta_\alpha^{j,a}, \quad a = 1,\dots,j, \tag{8}$$

where $a$ labels the individual rapidities within the $j$-string, while $\alpha$ labels different strings of length $j$. Here $\lambda_\alpha^j$ is a real number called the string centre. The structure (8) is common to many integrable systems and within the so called string hypothesis [75,77] the deviations from a perfect string $\delta_\alpha^{j,a}$ are assumed to be exponentially vanishing with the system size $L$ (see Refs. [78,79] for a numerical study of such deviations in the Lieb-Liniger model). A $j$-string can be seen to correspond to a bound state of $j$ bosons: indeed, one can show that the Bethe ansatz wave function decays exponentially with respect to the distance between any two particles in the bound state and the $j$ bosons can be thought as clustered together.

Even though some cases are known where states violating the string hypothesis are present [80–84], it is widely believed that their contribution to physically relevant quantities is vanishing in the thermodynamic limit. We will then always assume the deviations $\delta_\alpha^{j,a}$ to be exponentially small in $L$ and neglect them except when explicitly said otherwise.

From (6), (8) a system of equations for the string centres $\lambda_\alpha^j$ is obtained [76]

$$j\lambda_\alpha^j L - \sum_{(k,\beta)} \Phi_{jk}(\lambda_\alpha^j - \lambda_\beta^k) = 2\pi I_\alpha^j, \tag{9}$$

where

$$\Phi_{jk}(\lambda) = (1-\delta_{jk})\phi_{|j-k|}(\lambda) + 2\phi_{|j-k|+2}(\lambda) + \dots + 2\phi_{j+k-2}(\lambda) + \phi_{j+k}(\lambda), \tag{10}$$

$$\phi_j(\lambda) = 2\arctan\left(\frac{2\lambda}{j\bar{c}}\right), \tag{11}$$

and where $I_\alpha^j$ are integer (half-odd integer) for $N$ odd (even). Eqns (9) are called Bethe-Takahashi equations [75,85]. The momentum and the energy of a general eigenstate are then given by

$$K = \sum_{(j,\alpha)} j\lambda_\alpha^j, \qquad E = \sum_{(j,\alpha)} j(\lambda_\alpha^j)^2 - \frac{\bar{c}^2}{12}j(j^2-1). \tag{12}$$

## 2.2 The thermodynamic limit

In the repulsive case the thermodynamic limit

$$N, L \to \infty, \quad D = \frac{N}{L} \text{ fixed}, \tag{13}$$

was first considered in Ref. [86], and it is well studied in the literature. In the attractive case, the absolute value of the ground state energy in not extensive, but instead grows as $N^3$ [87,88]. While ground state correlation functions can be studied in the zero density limit, namely $N$ fixed, $L \to \infty$ [76], it was argued that the model does not have a proper thermodynamic limit in thermal equilibrium [75,88]. Crucially, in the quench protocol we are considering, the energy is fixed by the initial state and the limit of an infinite number of particles at fixed density presents no problem.

As the systems size $L$ grows, the centres of the strings associated with an energy eigenstate become a dense set on the real line and in the thermodynamic limit are described by smooth distribution function. In complete analogy with the standard finite-temperature formalism [75] we introduce the distribution function $\{\rho_n(\lambda)\}_{n=1}^\infty$ describing the centres of $n$ strings, and the distribution function of holes $\{\rho_n^h(\lambda)\}_{n=1}^\infty$. We recall that $\rho_n^h(\lambda)$ describes the distribution

of unoccupied states for the centres of $n$-particle strings, and is analogous to the distribution of holes in the case of ideal Fermi gases at finite temperature. Following Takahashi [75] we introduce

$$\eta_n(\lambda) = \frac{\rho_n^h(\lambda)}{\rho_n(\lambda)}, \tag{14}$$

$$\rho_n^t(\lambda) = \rho_n(\lambda) + \rho_n^h(\lambda). \tag{15}$$

In the thermodynamic limit the Bethe-Takahashi equations (9) reduce to an infinite set of coupled, non-linear integral equations

$$\frac{n}{2\pi} - \sum_{m=1}^{\infty} \int_{-\infty}^{\infty} d\lambda' a_{nm}(\lambda - \lambda') \rho_m(\lambda') = \rho_n(\lambda)(1 + \eta_n(\lambda)). \tag{16}$$

where

$$a_{nm}(\lambda) = (1 - \delta_{nm}) a_{|n-m|}(\lambda) + 2a_{|n-m|+2}(\lambda) + \ldots + 2a_{n+m-2}(\lambda) + a_{n+m}(\lambda), \tag{17}$$

$$a_n(\lambda) = \frac{1}{2\pi} \frac{d}{d\lambda} \phi_n(\lambda) = \frac{2}{\pi n \bar{c}} \frac{1}{1 + \left(\frac{2\lambda}{n\bar{c}}\right)^2}. \tag{18}$$

In the thermodynamic limit the energy and momentum per volume are given by

$$k[\{\rho_n\}] = \sum_{n=1}^{\infty} \int_{-\infty}^{\infty} d\lambda \, \rho_n(\lambda) n\lambda, \qquad e[\{\rho_n\}] = \sum_{n=1}^{\infty} \int_{-\infty}^{\infty} d\lambda \, \rho_n(\lambda) \varepsilon_n(\lambda), \tag{19}$$

where

$$\varepsilon_n(\lambda) = n\lambda^2 - \frac{\bar{c}^2}{12} n(n^2 - 1). \tag{20}$$

Finally, it is also useful to define the densities $D_n$ and energy densities $e_n$ of particles forming $n$-strings

$$D_n = n \int_{-\infty}^{\infty} d\lambda \, \rho_n(\lambda), \qquad e_n = \int_{-\infty}^{\infty} d\lambda \, \rho_n(\lambda) \varepsilon_n(\lambda). \tag{21}$$

The total density and energy per volume are then additive

$$D = \sum_{n=1}^{\infty} D_n, \qquad e = \sum_{n=1}^{\infty} e_n. \tag{22}$$

## 2.3 The quench protocol

We consider a quantum quench in which the system is initially prepared in the BEC state, i.e. the ground state of (1) with $c = 0$, and the subsequent unitary time evolution is governed by the Hamiltonian (1) with $c = -\bar{c} < 0$. The same initial state was considered for quenches to the repulsive Bose gas in Refs [29, 51, 52, 89–91], while different initial conditions were considered in Refs [55, 92–102].

As we mentioned before, the energy after the quench is conserved and is most easily computed in the initial state $|\psi(0)\rangle = |BEC\rangle$ as

$$\langle BEC | H_{LL} | BEC \rangle = -\bar{c} \langle BEC | \int_0^L dx \, \Psi^\dagger(x) \Psi^\dagger(x) \Psi(x) \Psi(x) | BEC \rangle. \tag{23}$$

The expectation value on the r.h.s. can then be easily computed using Wick's theorem. In the thermodynamic limit we have

$$\frac{E}{L} = -\bar{c} D^2 = -\gamma D^3. \tag{24}$$

## 3 The quench action method

### 3.1 General considerations

Consider the post-quench time evolution of the expectation value of a general operator $O$. For a generic system it can be written as

$$\langle\psi(t)|O|\psi(t)\rangle = \sum_{\mu,\nu}\langle\psi(0)|\mu\rangle\langle\mu|O|\nu\rangle\langle\nu|\psi(0)\rangle e^{i(E_\mu - E_\nu)t}, \tag{25}$$

where $\{|\mu\rangle\}$ denotes an orthonormal basis of eigenstates of the post-quench Hamiltonian. In Ref. [50] it was argued that in integrable systems a major simplification occurs if one is interested in the time evolution of the expectation values of *local* operators $\mathcal{O}$ in the thermodynamic limit. In particular, the double sum in the spectral representation (25) can be replaced by a single sum over particle-hole excitations over a *representative eigenstate* $|\rho_{sp}\rangle$. In particular, we have

$$\lim_{\text{th}}\langle\psi(t)|\mathcal{O}|\psi(t)\rangle = \frac{1}{2}\sum_{\mathbf{e}}\left(e^{-\delta s_{\mathbf{e}} - i\delta\omega_{\mathbf{e}}t}\langle\rho_{sp}|\mathcal{O}|\rho_{sp},\mathbf{e}\rangle + e^{-\delta s_{\mathbf{e}}^* + i\delta\omega_{\mathbf{e}}t}\langle\rho_{sp},\mathbf{e}|\mathcal{O}|\rho_{sp}\rangle\right), \tag{26}$$

where we have indicated with $\lim_{\text{th}}$ the thermodynamic limit $N, L \to \infty$, keeping the density $D = N/L$ fixed. Here $\mathbf{e}$ denotes a generic excitation over the representative state $|\rho_{sp}\rangle$. Finally we have

$$\delta s_{\mathbf{e}} = -\ln\frac{\langle\rho_{sp},\mathbf{e}|\psi(0)\rangle}{\langle\rho_{sp}|\psi(0)\rangle}, \qquad \delta\omega_{\mathbf{e}} = \omega[\rho_{sp},\mathbf{e}] - \omega[\rho_{sp}], \tag{27}$$

where $\omega[\rho_{sp}]$, $\omega[\rho_{sp},\mathbf{e}]$ are the energies of $|\rho_{sp}\rangle$ and $|\rho_{sp},\mathbf{e}\rangle$ respectively. The representative eigenstate (or "saddle-point state") $|\rho_{sp}\rangle$ is described in the thermodynamic limit by two sets of distribution functions $\{\rho_n(\lambda)\}_n$, $\{\rho_n^h(\lambda)\}_n$. In Ref. [50] it was argued that these are selected by the saddle-point condition

$$\left.\frac{\partial S_{QA}[\rho]}{\partial\rho_n(\lambda)}\right|_{\rho=\rho_{sp}} = 0, \qquad n \geq 1, \tag{28}$$

where $S_{QA}[\rho]$ is the so-called quench action

$$S_{QA}[\rho] = 2S[\rho] - S_{YY}[\rho]. \tag{29}$$

Here $\rho$ is the set of distribution functions corresponding to a general macro-state, $S[\rho]$ gives the thermodynamically leading part of the logarithm of the overlap

$$S[\rho] = -\lim_{\text{th}}\text{Re}\ln\langle\psi(0)|\rho\rangle, \tag{30}$$

and $S_{YY}$ is the Yang-Yang entropy. As we will see in section 3.2, we will only have to consider parity-invariant Bethe states, namely eigenstates of the Hamiltonian (1) characterised by sets of rapidities satisfying $\{\lambda_j\}_{j=1}^N = \{-\lambda_j\}_{j=1}^N$. Restricting to the sector of the Hilbert space of parity invariant Bethe states, the Yang-Yang entropy reads

$$\frac{S_{YY}[\rho]}{L} = \frac{1}{2}\sum_{n=1}^{\infty}\int_{-\infty}^{\infty}d\lambda[\rho_n\ln(1+\eta_n) + \rho_n^h\ln(1+\eta_n^{-1})]. \tag{31}$$

We note the global pre-factor $1/2$. From Eq. (26) it follows that the saddle-point state $|\rho_{sp}\rangle$ can be seen as the effective stationary state reached by the system at long times. Indeed, if $\mathcal{O}$ is a local operator, Eq. (26) gives

$$\lim_{t\to\infty}\lim_{\text{th}}\langle\psi(t)|\mathcal{O}|\psi(t)\rangle = \langle\rho_{sp}|\mathcal{O}|\rho_{sp}\rangle. \tag{32}$$

## 3.2 Overlaps with the BEC state

The main difficulty in applying the quench action method to a generic quantum quench problems is the computation of the overlaps $\langle\psi(0)|\rho\rangle$ between the initial state and eigenstates of the post-quench Hamiltonian. At present this problem has been solved only in a small number of special cases [50, 103–111].

A conjecture for the overlaps between the BEC state and the Bethe states in the Lieb-Liniger model first appeared in Ref. [51] and it was then rigorously proven, for arbitrary sign of the particle interaction strength, in Ref. [106]. As we have already mentioned, the overlap is non-vanishing only for parity invariant Bethe states, namely eigenstates characterised by sets of rapidities satisfying $\{\lambda_j\}_{j=1}^N = \{-\lambda_j\}_{j=1}^N$ [105]. The formula reads

$$\langle\{\lambda_j\}_{j=1}^{N/2} \cup \{-\lambda_j\}_{j=1}^{N/2}|\text{BEC}\rangle = \frac{\sqrt{(cL)^{-N}N!}}{\prod_{j=1}^{N/2}\frac{\lambda_j}{c}\sqrt{\frac{\lambda_j^2}{c^2}+\frac{1}{4}}}\frac{\det_{j,k=1}^{N/2}G_{jk}^Q}{\sqrt{\det_{j,k=1}^N G_{jk}}},\tag{33}$$

where

$$G_{jk} = \delta_{jk}\left[L + \sum_{l=1}^N K(\lambda_j - \lambda_l)\right] - K(\lambda_j - \lambda_k),\tag{34}$$

$$G_{jk}^Q = \delta_{jk}\left[L + \sum_{l=1}^{N/2} K^Q(\lambda_j, \lambda_l)\right] - K^Q(\lambda_j, \lambda_k),\tag{35}$$

$$K^Q(\lambda, \mu) = K(\lambda - \mu) + K(\lambda + \mu), \qquad K(\lambda) = \frac{2c}{\lambda^2 + c^2}.\tag{36}$$

The extensive part of the logarithm of the overlap (33) was computed in Ref. [51] in the repulsive regime. A key observation was that the ratio of the determinants is non-extensive, i.e.

$$\lim_{\text{th}}\frac{\det_{j,k=1}^{N/2}G_{jk}^Q}{\sqrt{\det_{j,k=1}^N G_{jk}}} = \mathcal{O}(1).\tag{37}$$

In the attractive regime additional technical difficulties arise, because matrix elements of the Gaudin-like matrices $G_{jk}$, $G_{jk}^Q$ can exhibit singularities when the Bethe state contains bound states [111]. This is analogous to the situation encountered for a quench from the Néel state to the gapped XXZ model [57–60]. In particular, one can see that the kernel $K(\mu - \nu)$ diverges as $1/(\delta_\alpha^{n,a} - \delta_\alpha^{n,a+1})$ for two "neighboring" rapidities in the same string $\mu = \lambda_\alpha^{n,a}$, $\nu = \lambda_\alpha^{n,a+1}$, or when rapidities from different strings approach one another in the thermodynamic limit, $\mu \to \lambda + ic$.

These kinds of singularities are present in the determinants of both $G_{jk}^Q$ and $G_{jk}$. It was argued in Refs [57, 58, 111] that they cancel one another in the expression for the overlap. As was noted in Refs. [57, 58, 111], no other singularities arise as long as one considers the overlap between the BEC state and a Bethe state without zero-momentum $n$-strings, (strings centred at $\lambda = 0$). Concomitantly the ratio of the determinants in (33) is expected to give a sub-leading contribution in the thermodynamic limit, and can be dropped. The leading term in the logarithm of the overlaps can then be easily computed along the lines of Refs. [57, 58]

$$\ln\langle\rho|\text{BEC}\rangle = -\frac{LD}{2}(\ln\gamma + 1) + \frac{L}{2}\sum_{m=1}^\infty \int_0^\infty d\lambda\,\rho_n(\lambda)\ln W_n(\lambda),\tag{38}$$

where

$$W_n(\lambda) = \frac{1}{\frac{\lambda^2}{c^2}\left(\frac{\lambda^2}{c^2}+\frac{n^2}{4}\right)\prod_{j=1}^{n-1}\left(\frac{\lambda^2}{c^2}+\frac{j^2}{4}\right)^2}.\tag{39}$$

In the case where zero-momentum $n$-strings are present, a more careful analysis is required in order to extract the leading term of the overlap (33) [111, 112]. This is reported in Appendix A. The upshot of this analysis is that (38) gives the correct leading behaviour of the overlap even in the presence of zero-momentum $n$-strings.

# 4 Stationary state

## 4.1 Saddle point equations

As noted before, the stationary state is characterized by two sets of distribution functions $\{\rho_n(\lambda)\}_n$, $\{\rho_n^h(\lambda)\}_n$, which fulfil two infinite systems of coupled, non-linear integral equations. The first of these is the thermodynamic version of the Bethe-Takahashi equations (16). The second set is derived from the saddle-point condition of the quench action (28), and the resulting equations are sometimes called the overlap thermodynamic Bethe ansatz equations (oTBA equations). Their derivation follows Refs [57–60]. In order to fix the density $D = N/L$ we add the following term to the quench action (29)

$$-hL\left(\sum_{m=1}^{\infty} m \int_{-\infty}^{\infty} d\lambda \rho_m(\lambda) - D\right). \tag{40}$$

As discussed in the previous section, $S[\rho]$ in (29) can be written as

$$S[\rho] = \frac{LD}{2}(\ln\gamma + 1) - \frac{L}{2}\sum_{m=1}^{\infty}\int_0^{\infty} d\lambda \rho_n(\lambda) \ln W_n(\lambda), \tag{41}$$

where $W_n(\lambda)$ is given in (39). Using (41), (31), and (40) one can straightforwardly extremize the quench action (29) and arrive at the following set of oTBA equations

$$\ln\eta_n(\lambda) = -2hn - \ln W_n(\lambda) + \sum_{m=1}^{\infty} a_{nm} * \ln\left(1 + \eta_m^{-1}\right)(\lambda), \qquad n \geq 1. \tag{42}$$

Here $a_{nm}$ are defined in (17), and we have used the notation $f * g(\lambda)$ to indicate the convolution between two functions

$$f * g(\lambda) = \int_{-\infty}^{\infty} d\mu \, f(\lambda - \mu)g(\mu). \tag{43}$$

Eqns (42) determine the functions $\eta_n(\lambda)$ and, together with Eqns (16) completely fix the distribution functions $\{\rho_n(\lambda)\}_n$, $\{\rho_n^h(\lambda)\}_n$ characterising the stationary state.

## 4.2 Tri-diagonal form of the oTBA equations

Following standard manipulations of equilibrium TBA equations [75], we may re-cast the oTBA equations (42) in the form

$$\ln\eta_n(\lambda) = d(\lambda) + s * [\ln(1 + \eta_{n-1}) + \ln(1 + \eta_{n+1})](\lambda), \qquad n \geq 1. \tag{44}$$

Here we have defined $\eta_0(\lambda) = 0$ and

$$s(\lambda) = \frac{1}{2\bar{c}\cosh\left(\frac{\pi\lambda}{\bar{c}}\right)}, \tag{45}$$

$$d(\lambda) = \ln\left[\tanh^2\left(\frac{\pi\lambda}{2\bar{c}}\right)\right]. \tag{46}$$

The calculations leading to Eqns (44) are summarized in Appendix B. The thermodynamic form of the Bethe-Takahashi equations (16) can be similarly rewritten. Since we do not make explicit use of them in the following, we relegate their derivation to Appendix B.

### 4.3 Asymptotic relations

Eqns (44) do not fix $\{\eta_n(\lambda)\}_n$ of Eqns (42), because they do not contain the chemical potential $h$. In order to recover the (unique) solution of Eqns (42), it is then necessary to combine Eqns (44) with a condition on the asymptotic behaviour of $\eta_n(\lambda)$ for large $n$. In our case one can derive from (42) the following relation, which holds asymptotically for $n \to \infty$

$$\ln \eta_{n+1}(\lambda) \simeq -2h + a_1 * \ln \eta_n(\lambda) + \ln\left[\frac{\lambda}{\bar{c}}\left(\frac{\lambda^2}{\bar{c}^2} + \frac{1}{4}\right)\right]. \tag{47}$$

Here $a_1(\lambda)$ is given in (18) (for $n = 1$). The derivation of Eqn (47) is reported in Appendix C. The set of equations (44), with the additional constraint given by Eqn (47), is now equivalent to Eqns (42).

## 5 Rapidity distribution functions for the stationary state

### 5.1 Numerical analysis

Eqns (16), (42) can be truncated to obtain a finite system of integral equations, which are defined on the real line $\lambda \in (-\infty, \infty)$. One can then numerically solve this finite system either by introducing a cut-off for large $\lambda$, or by mapping the equations onto a finite interval. Following the latter approach, we define

$$\chi_n(\lambda) = \ln\left(\frac{\eta_n(\lambda)\tau^{2n}}{q_n(\lambda)}\right), \tag{48}$$

where $q_n(\lambda)$ is given by

$$q_n(\lambda) = \frac{1}{W_n(\lambda)} = \frac{\lambda^2}{\bar{c}^2}\left(\frac{\lambda^2}{\bar{c}^2} + \left(\frac{n}{2}\right)^2\right)\prod_{l=1}^{n-1}\left[\frac{\lambda^2}{\bar{c}^2} + \left(\frac{l}{2}\right)^2\right]^2. \tag{49}$$

Finally, we have defined

$$\tau = e^h, \tag{50}$$

$h$ being the Lagrange multiplier appearing in (42). The functions $\chi_n(\lambda)$ satisfy the following system of equations

$$\begin{aligned}
\chi_n(\lambda) &= \sum_{m=1}^{\infty} a_{nm} * \ln\left(1 + \frac{\tau^{2m}}{q_m(\lambda)}e^{-\chi_m(\lambda)}\right) = \\
&= \sum_{m=1}^{\infty} \int_0^{+\infty} d\mu\,(a_{nm}(\lambda - \mu) + a_{nm}(\lambda + \mu))\ln\left(1 + \frac{\tau^{2m}}{q_m(\mu)}e^{-\chi_m(\mu)}\right),
\end{aligned} \tag{51}$$

where $a_{nm}(\lambda)$ are defined in (17). We then change variables

$$\frac{\lambda(x)}{\bar{c}} = \frac{1-x}{1+x}, \tag{52}$$

which maps the interval $(0, \infty)$ onto $(-1, 1)$. Since the distributions $\chi_n(\lambda)$ are symmetric w.r.t. 0, they can be described by functions with domain $(0, \infty)$. Using the map (52) they become functions $\chi_n(x)$ with domain $(-1, 1)$. The set of equations (51) becomes

$$\chi_n(x) = 2\sum_{m=1}^{\infty} \int_{-1}^{1} dy\,\frac{1}{(1+y)^2}\mathscr{A}_{nm}(x, y)\ln\left(1 + \frac{\tau^{2m}}{q_m(y)}e^{-\chi_m(y)}\right), \tag{53}$$

where

$$\mathscr{A}_{nm}(x,y) = \bar{c}\left[a_{nm}\Big(\lambda(x)-\lambda(y)\Big) + a_{nm}\Big(\lambda(x)+\lambda(y)\Big)\right]. \tag{54}$$

The thermodynamic Bethe-Takahashi equations (16) can be similarly recast in the form

$$\Theta_n(x) = \frac{n}{2\pi} - 2\sum_{m=1}^{\infty}\int_{-1}^{1}\frac{\mathrm{d}y}{(1+y)^2}\frac{\mathscr{A}_{nm}(x,y)}{1+\eta_m(y)}\Theta_m(y), \tag{55}$$

where $\Theta(x) = \rho_n^t\big(\lambda(x)\big)$, with $\lambda(x)$ defined in Eq. (52). The infinite systems (53) and (55), defined on the interval $(-1,1)$, can then be truncated and solved numerically for the functions $\chi_n(x)$ and $\Theta_n(x)$, for example using the Gaussian quadrature method. The functions $\eta_n(\lambda)$ are recovered from (48) and (52), while the particle and hole distributions $\rho_n(\lambda)$, $\rho_n^h(\lambda)$ are obtained from the knowledge of $\eta_n(\lambda)$ and $\rho_n^t(\lambda)$.

As $\gamma$ decreases, we found that an increasing number of equations has to be kept when truncating the infinite systems (53), (55) in order to obtain an accurate numerical solution. As we will see in section 6.2, this is due to the fact that, as $\gamma \to 0$, bound states with higher number of particles are formed and the corresponding distribution functions $\rho_n(\lambda)$, $\eta_n(\lambda)$ cannot be neglected in (16), (42). As an example, our numerical solution for $\gamma = 0.25$, and $\gamma = 2.5$ is shown in Fig. 1, where we also provide a comparison with the analytical solution discussed in section 5.3.

Two non-trivial checks for our numerical solution are available. The first is given by Eq. (24), i.e. the solution has to satisfy the sum rule

$$\sum_{n=1}^{\infty}\int_{-\infty}^{\infty}\mathrm{d}\lambda \rho_n(\lambda)\varepsilon_n(\lambda) = -\gamma D^3, \tag{56}$$

where $\varepsilon_n(\lambda)$ is defined in Eqn (20). The second non-trivial check was suggested in Refs [59,60] (see also Ref. [58]), and is based on the observation that the action (29) has to be equal to zero when evaluated on the saddle point solution, i.e. $S_{QA}[\rho_{sp}] = 0$, or

$$2S[\rho_{sp}] = S_{YY}[\rho_{sp}], \tag{57}$$

where $S[\rho]$ and $S_{YY}[\rho]$ are defined respectively in (41) and (31). Both (56) and (57) are satisfied by our numerical solutions within a relative numerical error $\epsilon \lesssim 10^{-4}$ for all numerically accessible values of $h$. As a final check we have verified that our numerical solution satisfies, within numerical errors,

$$\gamma = \frac{1}{\tau}, \tag{58}$$

where $\tau$ is defined in (50) and $\gamma = \bar{c}/D$ is computed from the distribution functions using (22). Relation (58) is equivalent to that found in the repulsive case [51].

## 5.2 Perturbative expansion

Following Ref. [51] we now turn to a "perturbative" analysis of Eqns (42). This will provide us with another non-trivial check on the validity of the analytical solution presented in section 5.3. Defining $\varphi_n(\lambda) = 1/\eta_n(\lambda)$ and using (50), we can rewrite (42) in the form

$$\ln\varphi_n(\lambda) = \ln(\tau^{2n}) + \ln W_n(\lambda) - \sum_{m=1}^{\infty}a_{nm}*\ln(1+\varphi_m)(\lambda), \tag{59}$$

where $W_n(\lambda)$ is given in (39). We now expand the functions $\varphi_n(\lambda)$ as power series in $\tau$

$$\varphi_n(\lambda) = \sum_{k=0}^{\infty}\varphi_n^{(k)}(\lambda)\tau^k. \tag{60}$$

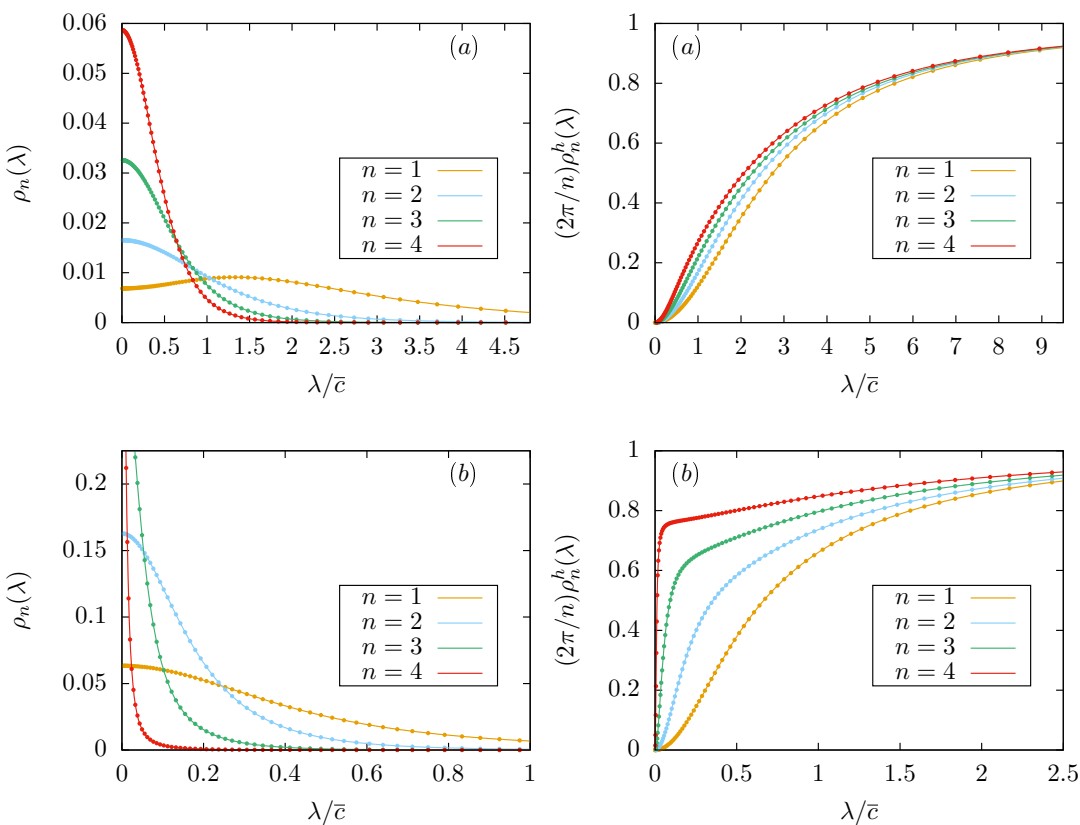

Figure 1: Rapidity distribution functions $\rho_n(\lambda)$ and $(2\pi/n)\rho_n^h(\lambda)$ for $n$-string solutions with $n \leq 4$. The final value of the interaction is chosen as (a) $\gamma = 0.25$ and (b) $\gamma = 2.5$. The dots correspond to the numerical solution discussed in section 5.1, while solid lines correspond to the analytical solution presented in section 5.3. The functions are shown for $\lambda > 0$ (being symmetric with respect to $\lambda = 0$) and have been rescaled for presentational purposes. Note that the rescaling factors for the hole distributions are determined by their asymptotic values, $\rho_n^h(\lambda) \rightarrow n/2\pi$ as $\lambda \rightarrow \infty$.

From (59) one readily sees that $\varphi_n(\lambda) = \mathcal{O}(\tau^{2n})$, i.e.

$$\varphi_n^{(k)}(\lambda) = 0, \quad k = 0, \ldots, 2n-1, \tag{61}$$

$$\varphi_n^{(2n)}(\lambda) = \frac{1}{\frac{\lambda^2}{\bar{c}^2}\left(\frac{\lambda^2}{\bar{c}^2} + \frac{n^2}{4}\right)\prod_{j=1}^{n-1}\left(\frac{\lambda^2}{\bar{c}^2} + \frac{j^2}{4}\right)^2}. \tag{62}$$

Using (62) as a starting point we can now solve Eqns (59) by iteration. The calculations are straightforward but tedious, and are sketched in Appendix D. Using this method we have calculated $\varphi_1(\lambda)$ up to fifth order in $\tau$. In terms of the the dimensionless variable $x = \lambda/\bar{c}$ we have

$$\varphi_1(x) = \frac{\tau^2}{x^2(x^2 + \frac{1}{4})}\left[1 - \frac{4\tau}{x^2 + 1} + \frac{\tau^2(1 + 13x^2)}{(1 + x^2)^2(x^2 + \frac{1}{4})} - \frac{32(-1 + 5x^2)\tau^3}{(1 + x^2)^3(1 + 4x^2)}\right] + \mathcal{O}(\tau^6). \tag{63}$$

## 5.3 Exact solution

In this section we discuss how to solve equations (16), (42) analytically. We first observe that the distribution functions $\rho_n(\lambda)$ can be obtained from the set $\{\eta_n(\lambda)\}_n$ of functions fulfilling Eqns (42) as

$$\rho_n(\lambda) = \frac{\tau}{4\pi}\frac{\partial_\tau \eta_n^{-1}(\lambda)}{1 + \eta_n^{-1}(\lambda)}, \tag{64}$$

where $\tau$ is given in (50). This relation is analogous to the one found in the repulsive case in Ref. [51]. To prove (64) one takes the partial derivative $\partial_\tau$ of both sides of (59). Combining the resulting equation with the thermodynamic version of the Bethe-Takahashi equations (16), and finally invoking the uniqueness of the solution, we obtain (64).

This leaves us with the task of solving (42). In what follows we introduce the dimensionless parameter $x = \lambda/\bar{c}$ and throughout this section, with a slight abuse of notation, we will use the same notations for functions of $\lambda$ and of $x$. Our starting point is the tri-diagonal form (44) of the coupled integral equations (42). Following Ref. [58] we introduce the corresponding $Y$-system [113, 114]

$$y_n\left(x + \frac{i}{2}\right)y_n\left(x - \frac{i}{2}\right) = Y_{n-1}(x)Y_{n+1}(x), \qquad n \geq 1, \tag{65}$$

where we define $y_0(x) = 0$ and

$$Y_n(x) = 1 + y_n(x). \tag{66}$$

Let us now assume that there exists a set of functions $y_n(x)$ that satisfy the $Y$-system (65), and as functions of the complex variable $z$ have the following properties

1. $y_n(z) \sim z^2$, as $z \to 0$, $\forall n \geq 1$;

2. $y_n(z)$ has no poles in $-1/2 < \text{Im}(z) < 1/2$, $\forall n \geq 1$;

3. $y_n(z)$ has no zeroes in $-1/2 < \text{Im}(z) < 1/2$ except for $z = 0$, $\forall n \geq 1$.

One can prove that the set of functions $y_n(x)$ with these properties solve the tri-diagonal form of the integral equations equations (44) [58]. To see this, one has to first take the logarithmic derivative of both sides of (65) and take the Fourier transform, integrating in $x \in (-\infty, \infty)$. Since the argument of the functions in the l.h.s. is shifted by $\pm i/2$ in the imaginary direction, one has to use complex analysis techniques to perform the integral. In particular, under the assumptions (1), (2), (3) the application of the residue theorem precisely generates, after taking the inverse Fourier transform, the driving term $d(\lambda)$ in (44) [58].

We conjecture that the exact solution for $\eta_1(x)$ is given by

$$\eta_1(x) = \frac{x^2[1 + 4\tau + 12\tau^2 + (5 + 16\tau)x^2 + 4x^4]}{4\tau^2(1 + x^2)}. \tag{67}$$

Our evidence supporting this conjecture is as follows:

1. We have verified using Mathematica that the functions $\eta_n(x)$ generated by substituting (67) into the Y-system (65) have the properties (1), (2) up to $n = 30$. We have checked for a substantial number of values of the chemical potential $h$ that they have the third property (3) up to $n = 10$.

2. Our expression (67) agrees with the expansion (63) in powers of $\tau$ up to fifth order.

3. Eqn (67) agrees perfectly with our numerical solution of the saddle-point equations discussed in section 5.1, as is shown in Fig. 1.

Given $\eta_1(x)$ we can use the $Y$-system (65) to generate $\eta_n(x)$ with $n \geq 2$

$$\eta_n(x) = \frac{\eta_{n-1}\left(x + \frac{i}{2}\right)\eta_{n-1}\left(x - \frac{i}{2}\right)}{1 + \eta_{n-2}(x)} - 1 , \ n \geq 2. \tag{68}$$

As mentioned before, the distribution functions $\rho_n(x)$ can be obtained using (64). The explicit expressions for $\rho_1(x)$ and $\rho_2(x)$ are as follows:

$$\rho_1(x) = \frac{2\tau^2(1 + x^2)(1 + 2\tau + x^2)}{\pi(x^2 + (2\tau + x^2)^2)(1 + 5x^2 + 4(\tau + 3\tau^2 + 4\tau x^2 + x^4))}, \tag{69}$$

$$\rho_2(x) = \frac{16\tau^4(9 + 4x^2)h_1(x, \tau)}{\pi(1 + 4x^2 + 8\tau)h_2(x, \tau)h_3(x, \tau)}, \tag{70}$$

where

$$
\begin{aligned}
h_1(x, \tau) &= 9 + 49x^2 + 56x^4 + 16x^6 + 72\tau \\
&+ 168x^2\tau + 96x^4\tau + 116\tau^2 + 176x^2\tau^2 + 96\tau^3, \\
h_2(x, \tau) &= 9 + 49x^2 + 56x^4 + 16x^6 + 24\tau \\
&+ 120x^2\tau + 96x^4\tau + 40\tau^2 + 160x^2\tau^2 + 64\tau^3, \\
h_3(x, \tau) &= 9x^2 + 49x^4 + 56x^6 + 16x^8 + 96x^2\tau + 224x^4\tau \\
&+ 128x^6\tau + 232x^2\tau^2 + 352x^4\tau^2 + 384x^2\tau^3 + 144\tau^4.
\end{aligned}
$$

$$\text{(71)}$$
$$\text{(72)}$$
$$\text{(73)}$$

The functions $\rho_n(x)$ for $n \geq 3$ are always written as rational functions but their expressions get lengthier as $n$ increases.

# 6 Physical properties of the stationary state

## 6.1 Local pair correlation function

The distribution functions $\rho_n(\lambda)$, $\rho_n^h(\lambda)$ completely characterize the stationary state. Their knowledge, in principle, allows one to calculate all local correlation functions in the thermodynamic limit. In practice, while formulas exist for the expectation values of simple local operators in the Lieb-Liniger model in the finite volume [115–118], it is generally difficult to take the thermodynamic limit of these expressions. In contrast to the repulsive case [117,119–123],

much less is known in the attractive regime, where technical complications arise that are associated with the existence of string solutions to the Bethe ansatz equations. Here we focus on the computation of the local pair correlation function

$$g_2 = \frac{\langle : \hat{\rho}^2(0) : \rangle}{D^2} = \frac{\langle \Psi^\dagger(0)\Psi^\dagger(0)\Psi(0)\Psi(0) \rangle}{D^2}. \tag{74}$$

We start by applying the Hellmann-Feynman [119, 120, 122, 124] theorem to the expectation value in a general energy eigenstate $|\{\lambda_j\}\rangle$ with energy $E[\{\lambda_j\}]$ of the finite system

$$\langle \{\lambda_j\}|\Psi^\dagger\Psi^\dagger\Psi\Psi|\{\lambda_j\}\rangle = -\frac{1}{L}\frac{\partial E[\{\lambda_j\}]}{\partial \bar{c}}. \tag{75}$$

In order to evaluate the expression on the r.h.s., we need to take the derivative of the Bethe-Takahashi equations (9) with respect to $\bar{c}$

$$f^{(n)}(\lambda_\alpha) = \frac{1}{n}\sum_m \frac{2\pi}{L}\sum_\beta \left( f^{(n)}(\lambda_\alpha) - f^{(m)}(\lambda_\beta) - \frac{\lambda_\alpha^n}{\bar{c}} + \frac{\lambda_\beta^m}{\bar{c}} \right) a_{nm}(\lambda_\alpha^n - \lambda_\beta^m). \tag{76}$$

Here $a_{nm}$ is given in Eq. (17) and

$$f^{(n)}(\lambda_\alpha) = \frac{\partial \lambda_\alpha^n}{\partial \bar{c}}. \tag{77}$$

Taking the thermodynamic limit gives

$$f^{(n)}(\lambda) = \frac{2\pi}{n}\left( f^{(n)}(\lambda) - \frac{\lambda}{\bar{c}} \right)\sum_{m=1}^\infty \int_{-\infty}^\infty d\mu\, \rho_m(\mu) a_{nm}(\lambda - \mu)$$

$$+ \frac{2\pi}{n}\sum_{m=1}^\infty \int_{-\infty}^\infty d\mu\, \rho_m(\mu)\left( \frac{\mu}{\bar{c}} - f^{(m)}(\mu) \right) a_{nm}(\lambda - \mu). \tag{78}$$

Using the thermodynamic version of the Bethe-Takahashi equations (16) and defining

$$b_n(\lambda) = 2\pi\left( \frac{\lambda}{\bar{c}} - f^{(n)}(\lambda) \right)\rho_n^t(\lambda), \tag{79}$$

we arrive at

$$b_n(\lambda) = n\frac{\lambda}{\bar{c}} - \sum_{m=1}^\infty \int_{-\infty}^\infty d\mu\, \frac{1}{1+\eta_m(\mu)}b_m(\mu) a_{nm}(\lambda - \mu). \tag{80}$$

The set of equations (80) completely fixes the functions $b_n(\lambda)$, once the functions $\eta_n(\lambda)$ are known. The right hand side of (75) in the finite volume can be cast in the form

$$\frac{\partial E}{\partial \bar{c}} = \sum_n \left[ \sum_\alpha 2n\lambda_\alpha^n f^{(n)}(\lambda_\alpha) - \frac{\bar{c}}{6}n(n^2 - 1) \right]. \tag{81}$$

Taking the thermodynamic limit, and using (79) we finally arrive at

$$\frac{1}{L}\frac{\partial E}{\partial \bar{c}} = \sum_{n=1}^\infty \int_{-\infty}^\infty \frac{d\mu}{2\pi}\left[ 2\pi\rho_n(\mu)\left( \frac{2n\mu^2}{\bar{c}} - \frac{\bar{c}}{6}n(n^2 - 1) \right) - 2n\mu b_n(\mu)\frac{1}{1+\eta_m(\mu)} \right]. \tag{82}$$

Combining (80) and (82) we can express the local pair correlation function as

$$g_2(\gamma) = \gamma^2 \sum_{m=1}^\infty \int_{-\infty}^\infty \frac{dx}{2\pi}\left[ 2mx\, b_m(x)\frac{1}{1+\tilde{\eta}_m(x)} - 2\pi\tilde{\rho}_m(x)\left( 2mx^2 - \frac{m(m^2-1)}{6} \right) \right], \tag{83}$$

where the functions $b_n(x)$ are determined by

$$b_n(x) = nx - \sum_{m=1}^{\infty} \int_{-\infty}^{\infty} dy \, \frac{1}{1 + \widetilde{\eta}_m(y)} b_m(y) \widetilde{a}_{nm}(x - y). \tag{84}$$

In (83), (84) we defined

$$\widetilde{\eta}_n(x) = \eta_n(x\overline{c}), \quad \widetilde{\rho}_n(x) = \rho_n(x\overline{c}), \quad \widetilde{a}_{nm}(x) = \overline{c} a_{nm}(x\overline{c}). \tag{85}$$

Using the knowledge of the functions $\eta_n(\lambda)$ for the stationary state, we can solve Eqns (84) numerically and substitute the results into (83) to obtain $g_2(\gamma)$.

While (83), (84) cannot be solved in closed form, they can be used to obtain an explicit asymptotic expansion around $\gamma = \infty$. To that end we use (19), (20) and (24) to rewrite $g_2(\gamma)$ as

$$g_2(\gamma) = 2 + \gamma^2 \sum_{m=1}^{\infty} \int_{-\infty}^{\infty} \frac{dx}{2\pi} 2mx b_m(x) \frac{1}{1 + \widetilde{\eta}_m(x)}. \tag{86}$$

We then use that large values of $\gamma$ correspond to small values of $\tau$, cf. (58), and carry out a small-$\tau$ expansion of the functions

$$\frac{1}{1 + \widetilde{\eta}_n(x)} = \frac{\widetilde{\varphi}_n(x)}{(1 + \widetilde{\varphi}_n(x))}, \tag{87}$$

where $\widetilde{\varphi}_n(x) = 1/\widetilde{\eta}_n(x)$ as in section 5.2. Substituting this expansion into the r.h.s. of (84) and proceeding iteratively, we obtain an expansion for the functions $b_n(x)$ in powers of $\tau$. The steps are completely analogous to those discussed in section 5.2 for the functions $\varphi_n(\lambda)$ and will not be repeated here. Finally, we use the series expansions of $b_n(x)$ and $(1 + \widetilde{\eta}_n(x))^{-1}$ in (86) to obtain an asymptotic expansion for $g_2(\gamma)$. The result is

$$g_2(\gamma) = 4 - \frac{40}{3\gamma} + \frac{344}{3\gamma^2} - \frac{2656}{3\gamma^3} + \frac{1447904}{225\gamma^4} + \mathcal{O}(\gamma^{-5}). \tag{88}$$

In Fig. 2 we compare results of a full numerical solution of Eqns (83), (84) to the asymptotic expansion (88). As expected, the latter breaks down for sufficiently small values of $\gamma$. In contrast to the large-$\gamma$ regime, the limit $\gamma \to 0$ is more difficult to analyze because $g_2(\gamma)$ is non-analytic in $\gamma = 0$. The limit $\gamma \to 0$ can be calculated as shown in Appendix E, and is given by

$$\lim_{\gamma \to 0} g_2(\gamma) = 2. \tag{89}$$

As was already noted in Ref. [56], (89) implies that the function $g_2(\gamma)$ is discontinuous in $\gamma = 0$. Indeed, $g_2(0)$ can be calculated directly by using Wick's theorem in the initial BEC state

$$\frac{\langle \text{BEC} | : \hat{\rho}(0)^2 : | \text{BEC} \rangle}{D^2} = 1. \tag{90}$$

This discontinuity, which is absent for quenches to the repulsive regime [51], is ascribed to the presence of multi-particle bound states for all values of $\gamma \neq 0$. The former are also at the origin of the non-vanishing limit of $g_2(\gamma)$ for $\gamma \to \infty$ as it will be discussed in the next section.

Finally, an interesting question is the calculation of the three-body one-point correlation function $g_3(\gamma)$ on the post-quench steady state. The latter is relevant for experimental realizations of bosons confined in one dimension, as it is proportional to the three-body recombination rate [125]. For $g_3$ it is reasonable to expect that three-particle bound states may give non-vanishing contributions in the large coupling limit. While $g_3$ is known for general states in the repulsive Lieb-Liniger model, its computation in the attractive case is significantly harder and requires further development of existing methods. We hope that our work will motivate theoretical efforts in this direction.

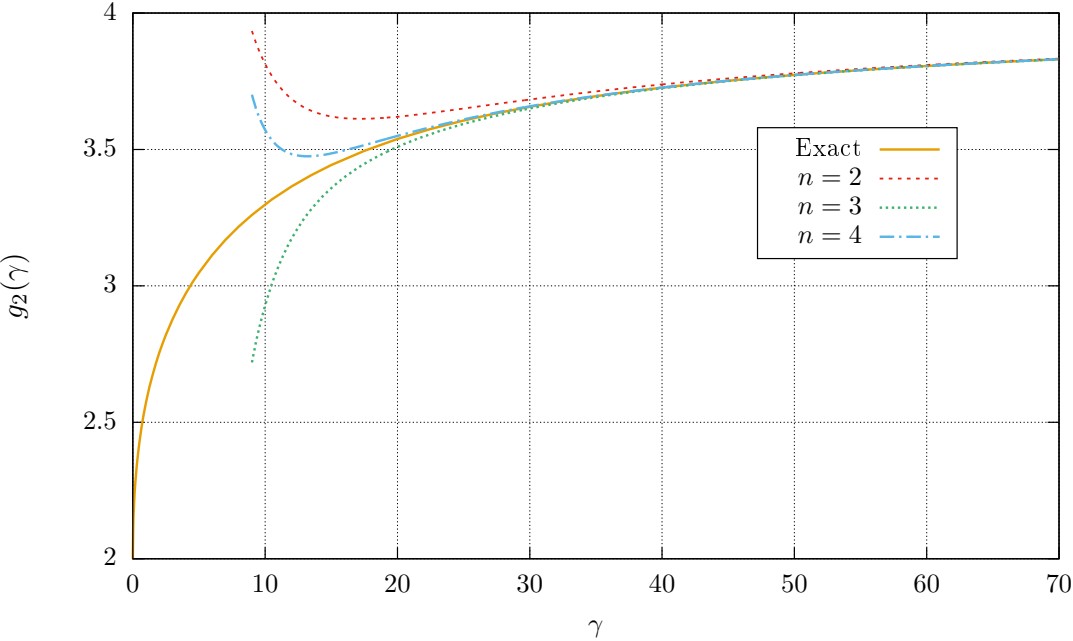

Figure 2: Local pair correlation function $g_2(\gamma)$ in the stationary state at late times after the quench. The numerical solution of Eqns (83), (84) is shown as a solid orange line. The asymptotic expansion (88) around $\gamma = \infty$ up to order $\mathcal{O}(\gamma^{-n})$ with $n = 2, 3, 4$ is seen to be in good agreement for large values of $\gamma$.

## 6.2 Physical implications of the multi-particle bound states

A particularly interesting feature of our stationary state is the presence of finite densities of $n$-particle bound states with $n \geq 2$. In Fig. 3, their densities and energies per volume are shown for a number of different values of $\gamma$. We see that the maximum of $D_n$ occurs at a value of $n$ that increases as $\gamma$ decreases. This result has a simple physical interpretation. In the attractive regime, the bosons have a tendency to form multi-particle bound states. One might naively expect that increasing the strength $\gamma$ of the attraction between bosons would lead to the formation of bound states with an ever increasing number of particles, thus leading to phase separation. However, in the quench setup the total energy of the system is fixed by the initial state, cf. (24), while the energy of $n$-particle bound states scales as $n^3$, cf. Eqns (20), (21). As a result, $n$-particle bound states cannot be formed for large values of $\gamma$, and indeed they are found to have very small densities for $n \geq 3$. On the contrary, decreasing the interaction strength $\gamma$, the absolute value of their energy lowers and these bound states become accessible. The dependence of the peak in Fig. 3 on $\gamma$ is monotonic but non-trivial and it is the result of the competition between the tendency of attractive bosons to cluster, and the fact that $n$-particle bound states with $n$ very large cannot be formed as a result of energy conservation.

The presence of multi-particle bound states affects measurable properties of the system, and is the reason for the particular behaviour of the local pair correlation function computed in the previous section. Remarkably, this is true also in the limit $\gamma \to \infty$. This is in marked contrast to the super Tonks-Girardeau gas, where bound states are absent. To exhibit the important role of bound states in the limit of large $\gamma$, we will demonstrate that the limiting value of $g_2(\gamma)$ for $\gamma \to \infty$ is entirely determined by bound pairs. It follows from (83) that

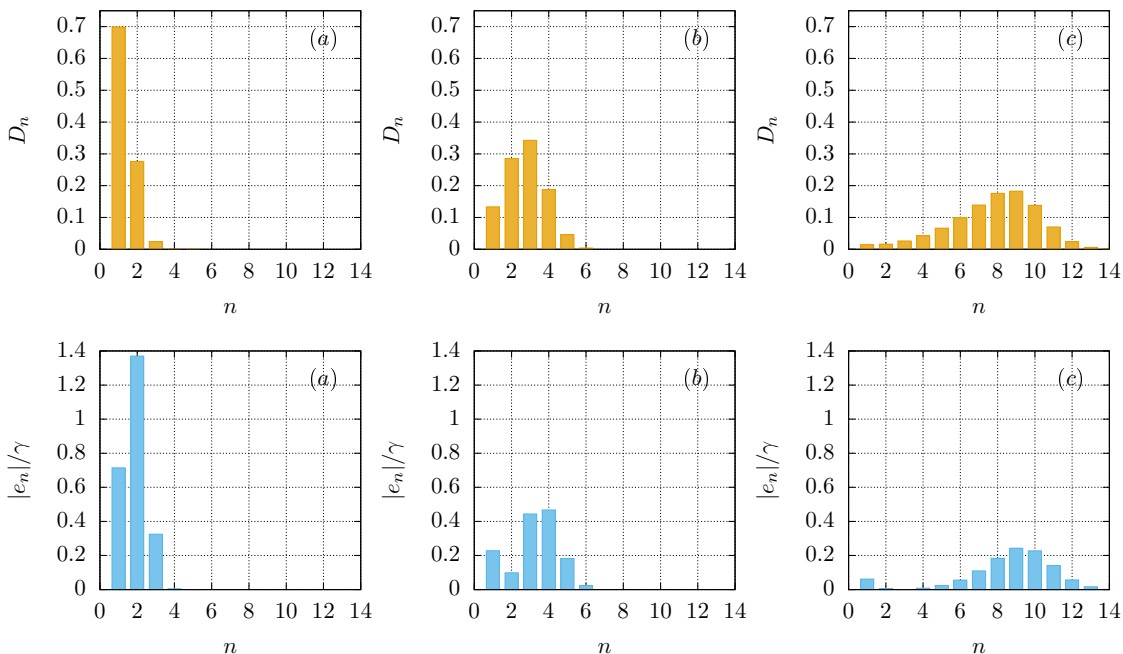

Figure 3: Density $D_n$ and absolute value of the normalized energies per volume $e_n/\gamma$ of the bosons forming $n$-particle bound states as defined in (21). The plots correspond to $(a)\ \gamma = 20$, $(b)\ \gamma = 2$, $(c)\ \gamma = 0.2$. The total density is fixed $D = 1$. The energy densities $e_n$ are always negative for $n \geq 2$ (i.e. $|e_n| = -e_n$ for $n \geq 2$) while $e_1 > 0$.

$g_2(\gamma)$ can be written in the form

$$g_2(\gamma) = \sum_{m=1}^{\infty} g_2^{(m)}(\gamma), \tag{91}$$

where $g_2^{(m)}(\gamma)$ denotes the contribution of $m$-particle bound states to the local pair correlation

$$g_2^{(m)}(\gamma) = \gamma^2 \int_{-\infty}^{\infty} \frac{dx}{2\pi} \left[ 2mx\, b_m(x) \frac{1}{1+\widetilde{\eta}_m(x)} - 2\pi \widetilde{\rho}_m(x) \left( 2mx^2 - \frac{m(m^2-1)}{6} \right) \right]. \tag{92}$$

Let us first show that unbound particles give a vanishing contribution

$$\lim_{\gamma \to \infty} g_2^{(1)}(\gamma) = 0. \tag{93}$$

In order to prove this, we use that at leading order in $1/\gamma$ we have $b_1(x) = x$. Using the explicit expressions for $\widetilde{\eta}_1(x)$, $\widetilde{\rho}_1(x)$ we can then perform the integrations in the r.h.s. of Eq. (92) exactly and take the limit $\gamma \to \infty$ afterwards. We obtain

$$\lim_{\gamma \to \infty} \gamma^2 \int_{-\infty}^{\infty} \frac{dx}{2\pi}\, 2x\, b_1(x) \frac{1}{1+\widetilde{\eta}_1(x)} = 2, \tag{94}$$

$$\lim_{\gamma \to \infty} \gamma^2 \int_{-\infty}^{\infty} \frac{dx}{2\pi} \left( -2\pi \widetilde{\rho}_1(x) 2x^2 \right) = -2, \tag{95}$$

which establishes (93). Next, we address the bound pair contribution. At leading order in $1/\gamma$ we have $b_2(x) = 2x$, and using the explicit expression for $\widetilde{\eta}_2(x)$ we obtain

$$\lim_{\gamma \to \infty} \gamma^2 \int_{-\infty}^{\infty} \frac{dx}{2\pi}\, 4x\, b_2(x) \frac{1}{1+\widetilde{\eta}_2(x)} = 0. \tag{96}$$

This leaves us with the contribution

$$\lim_{\gamma \to \infty} \gamma^2 \int_{-\infty}^{\infty} \frac{\mathrm{d}x}{2\pi} \left[ -2\pi \widetilde{\rho_2}(x)\left(4x^2 - 1\right) \right]. \tag{97}$$

Although the function $\widetilde{\rho}_2(x)$ is known, cf. Eq. (70), its expression is unwieldy and it is difficult to compute the integral analytically. On the other hand, one cannot expand $\widetilde{\rho}_2(x)$ in $1/\gamma$ inside the integral, because the integral of individual terms in this expansion are not convergent (signalling that in this case one cannot exchange the order of the limit $\gamma \to \infty$ and of the integration). Nevertheless, the numerical computation of the integral in (97) for large values of $\gamma$ presents no difficulties and one can then compute the limit numerically. We found that the limit in Eq. (97) is equal to 4 within machine precision so that

$$\lim_{\gamma \to \infty} g_2(\gamma) = 4 = \lim_{\gamma \to \infty} g_2^{(2)}(\gamma). \tag{98}$$

Finally, we verified that contributions coming from bound states with higher numbers of particles are vanishing, i.e. $g_2^{(m)}(\gamma) \to 0$ for $\gamma \to \infty$, $m \geq 3$. This establishes that the behaviour of $g_2(\gamma)$ for large values of $\gamma$ is dominated by bound pair of bosons.

## 7 Conclusions

We have considered quantum quenches from an ideal Bose condensate to the one-dimensional Lieb-Liniger model with arbitrary attractive interactions. We have determined the stationary state, and determined its physical properties. In particular, we revealed that the stationary state is composed of an interesting mixture of multi-particle bound states, and computed the local pair correlation function in this state. Our discussion presents a detailed derivation of results first announced in Ref. [56].

As we have stressed repeatedly, the most intriguing feature of the stationary state for the quench studied in this work is the presence of multi-particle bound states. As was argued in Ref. [56], their properties could in principle be probed in ultra-cold atoms experiments. Multi-particle bound states are also formed in the quench from the Néel state to the gapped XXZ model, as it was recently reported in Refs. [57–60]. However, in contrast to our case, the bound state densities are always small compared to the density of unbound magnons for all the values of the final anisotropy parameter $\Delta \geq 1$ [58].

Our work also provides an interesting physical example of a quantum quench, where different initial conditions lead to stationary states with qualitatively different features. Indeed, a quench in the one-dimensional Bose gas from the infinitely repulsive to the infinitely attractive regime leads to the super Tonks-Girardeau gas, where bound states are absent. On the other hand, as shown in section 6, if the initial state is an ideal Bose condensate, bound states have important consequences on the correlation functions of the system even in the limit of large negative interactions.

An interesting open question is to find a description of our stationary state in terms of a GGE. As the stationary state involves bound states, it is likely that the GGE will involve not yet known quasi-local conserved charges [41, 44, 45] as well as the known ultra-local ones [126]. In the Lieb-Liniger model technical difficulties arise when addressing such issues, as expectation values of local conserved charges generally exhibit divergences [29, 89, 126]. In addition, very little is known about quasi-local conserved charges for interacting models defined in the continuum [41, 48].

Finally, it would be interesting to investigate the approach to the steady state in the quench considered in this work. While this is in general a very difficult problem, in the repulsive

regime the post-quench time evolution from the non-interacting BEC state was considered in [53]. There an efficient numerical evaluation of the representation (26) was performed, based on the knowledge of exact one-point form factors [118]. The attractive regime, however, is significantly more involved due to the presence of bound states and the study of the whole post-quench time evolution remains a theoretical challenge for future investigations.

## Acknowledgements

We thank Michael Brockmann for a careful reading of the manuscript. PC acknowledges the financial support by the ERC under Starting Grant 279391 EDEQS. The work of FHLE was supported by the EPSRC under grant EP/N01930X. All authors acknowledge the hospitality of the Isaac Newton Institute for Mathematical Sciences under grant EP/K032208/1.

## A  Overlaps in the presence of zero-momentum $n$-strings

In this appendix we argue that Eq. (38) gives the leading term in the thermodynamic limit of the logarithm of the overlap between the BEC state and a parity-invariant Bethe state, even in cases where the latter contains zero-momentum strings.

To see this, consider a parity invariant Bethe state with a single zero-momentum $m$-string, and $K$ parity-related pairs of $n_j$-strings. The total number of particles in such a state is then $N = 2\sum_j n_j + m$. In Ref. [111] an explicit expression for the overlap (33) of such states with a BEC state in the zero-density limit ($L \to \infty$ and $N$ fixed) was obtained. Up to an irrelevant (for our purposes) overall minus sign, it reads

$$
\begin{aligned}
\langle \{\lambda_j\}_{j=1}^{N/2} \cup \{-\lambda_j\}_{j=1}^{N/2} | \text{BEC} \rangle &= \frac{2^{m-1} L\bar{c}}{(m-1)!} \sqrt{\frac{N!}{(L\bar{c})^N}} \\
&\times \prod_{p=1}^{K} \frac{1}{\sqrt{\frac{\lambda_p^2}{\bar{c}^2}\left(\frac{\lambda_p^2}{\bar{c}^2} + \frac{n_p^2}{4}\right)} \prod_{q=1}^{n_p-1}\left(\frac{\lambda_p^2}{\bar{c}^2} + \frac{q^2}{4}\right)},
\end{aligned}
\tag{99}
$$

where $\lambda_p$ is the centre of the $p$'th string. We see that as a result of having a zero-momentum string, an additional pre-factor $L$ appears. In general, the presence of $M$ zero-momentum strings will lead to an additional pre-factor $L^M$ [111]. While (99) is derived in the zero density limit, we expect an additional pre-factor to be present also if one considers the thermodynamic limit $N, L \to \infty$, at finite density $D = N/L$. Importantly such pre-factors will result in *sub-leading* corrections of order $(\ln L)/L$ to the logarithm of the overlaps. This suggests that (38) holds even for states with zero-momentum $n$-strings.

## B  Tri-diagonal form of the coupled integral equations

### B.1  Tri-diagonal Bethe-Takahashi equations

Our starting point are the thermodynamic Bethe equations (16). For later convenience we introduce the following notations for the Fourier transform of a function

$$
\hat{f}(k) = \mathscr{F}[f](k) = \int_{-\infty}^{\infty} f(\lambda) e^{ik\lambda} d\lambda ,
\tag{100}
$$

$$f(\lambda) = \mathscr{F}^{-1}[\hat{f}](\lambda) = \frac{1}{2\pi} \int_{-\infty}^{\infty} \hat{f}(k) e^{-ik\lambda} dk .\tag{101}$$

We recall that $f * g$ denotes the convolution of two functions, cf. (43). The Fourier transform of $a_n(\lambda)$ defined in (18) is easily computed

$$\hat{a}_n(k) = e^{-\frac{n\bar{c}|k|}{2}} .\tag{102}$$

Following Ref. [85], we introduce the symbols

$$[nmp] = \begin{cases} 1 , & \text{if } p = |m-n| \text{ or } m+n \\ 2 , & \text{if } p = |m-n|+2, |m-n|+4, \ldots, m+n-2 , \\ 0 & \text{otherwise .} \end{cases}\tag{103}$$

We can then perform the Fourier transform of both sides of (16) and obtain

$$n\delta(k) - \sum_{m=1} \sum_{p>0} [nmp]\hat{\rho}_m(k) e^{-\frac{\bar{c}}{2}|k|p} = \hat{\rho}_n^t(k) ,\tag{104}$$

where $\rho_n^t(\lambda)$ are given in (15). We now define

$$\hat{\rho}_{-m}(k) = -\hat{\rho}_m(k) , \qquad m \geq 1,\tag{105}$$
$$\hat{\rho}_0(k) = 0 .\tag{106}$$

After straightforward calculations, we can rewrite (104) in the form

$$\hat{\rho}_n^h(k) = n\delta(k) - \coth\left(\frac{|k|\bar{c}}{2}\right) \sum_{m=-\infty}^{+\infty} e^{-|k||n-m|\frac{\bar{c}}{2}} \hat{\rho}_m(k) .\tag{107}$$

In order to decouple these equations we note that

$$\begin{aligned}
\hat{\rho}_{n+1}^h(k) \quad + \quad & \hat{\rho}_{n-1}^h(k) = 2n\delta(k) \\
- \quad & \coth\left(\frac{|k|\bar{c}}{2}\right) \left[ -2\hat{\rho}_n(k)\sinh\left(\frac{|k|\bar{c}}{2}\right) + 2\cosh\left(\frac{|k|\bar{c}}{2}\right) \sum_{m=-\infty}^{\infty} e^{-|k||n-m|\frac{\bar{c}}{2}} \hat{\rho}_m(k) \right] .
\end{aligned}\tag{108}$$

Combining Eqns (107), (108) one obtains

$$\begin{aligned}
\hat{\rho}_n^t(k) &= \frac{1}{2\cosh(|k|\bar{c}/2)} \left(\hat{\rho}_{n+1}^h(k) + \hat{\rho}_{n-1}^h(k)\right) - n\delta(k) \underbrace{\left[ \frac{1 - \cosh\left(\frac{|k|\bar{c}}{2}\right)}{\cosh\left(\frac{|k|\bar{c}}{2}\right)} \right]}_{=0} = \\
&= \frac{1}{2\cosh(|k|\bar{c}/2)} \left(\hat{\rho}_{n+1}^h(k) + \hat{\rho}_{n-1}^h(k)\right) .
\end{aligned}\tag{109}$$

We can now perform the inverse Fourier transform. Using

$$\frac{1}{2\pi} \int_{-\infty}^{\infty} dk \frac{1}{\cosh\left(k\frac{\bar{c}}{2}\right)} e^{-i\lambda k} = \frac{1}{\bar{c}} \frac{1}{\cosh\left(\frac{\lambda\pi}{\bar{c}}\right)} ,\tag{110}$$

we finally obtain

$$\rho_n(1 + \eta_n) = s * (\eta_{n-1}\rho_{n-1} + \eta_{n+1}\rho_{n+1}) \qquad n \geq 1 ,\tag{111}$$

where we can choose $\eta_0(\lambda)\rho_0(\lambda) = \delta(\lambda)$, $\eta_n(\lambda)$ is given in Eq. (14), and where

$$s(\lambda) = \frac{1}{2\bar{c}\cosh\left(\frac{\pi\lambda}{\bar{c}}\right)} .\tag{112}$$

## B.2 Tri-diagonal oTBA equations

In this appendix we derive the tri-diagonal equations (44) starting from Eqns (42). Our discussion follows Ref. [57]. Some useful identities are [75]

$$(a_0 + a_2) * a_{nm} = a_1 * (a_{n-1,m} + a_{n+1,m}) + (\delta_{n-1,m} + \delta_{n+1,m}) a_1 , \qquad n \geq 2, \ m \geq 1, \quad (113)$$

$$(a_0 + a_2) * a_{1m} = a_1 * a_{2,m} + a_1 \delta_{2,m} , \qquad m \geq 1 , \quad (114)$$

where we define $a_0(\lambda) = \delta(\lambda)$, and where the functions $a_{nm}(\lambda)$, $a_n(\lambda)$ are given in Eqns (17), (18). Convolution of (42) with $(a_0 + a_2)$ gives

$$\begin{aligned}
(a_0 + a_2) * \ln \eta_n &= (a_0 + a_2) * g_n - a_1 * (g_{n-1} + g_{n+1}) \\
&+ a_1 * [\ln(1 + \eta_{n-1}) + \ln(1 + \eta_{n+1})] , \qquad n \geq 1 ,
\end{aligned} \quad (115)$$

where we defined $g_n(\lambda) = -\ln W_n(\lambda)$, $g_0(\lambda) = 0$ and $\eta_0(\lambda) = 0$. The functions $g_n(\lambda)$ can be written as

$$g_n(\lambda) = \ln s_0^{(2)}(\lambda) + \ln s_n^{(2)}(\lambda) + 2 \sum_{\ell=1}^{n-1} \ln s_\ell^{(2)}(\lambda) , \quad (116)$$

where

$$s_\ell^{(2)}(\lambda) = s_\ell(\lambda) s_{-\ell}(\lambda) = \frac{\lambda^2}{\bar{c}^2} + \frac{\ell^2}{4} . \quad (117)$$

It is straightforward to show that

$$(a_m * f_r)(\lambda) = f_{m+r}(\lambda), \quad (118)$$

where we defined

$$f_r(\lambda) = \ln \left[ \left( \frac{\lambda}{\bar{c}} \right)^2 + \left( \frac{r}{2} \right)^2 \right]. \quad (119)$$

Using (118) and (116), we can rewrite the driving term in (115) as

$$\tilde{d}_n \equiv (a_0 + a_2) * g_n - a_1 * (g_{n-1} + g_{n+1}) = f_0 - f_2 = \ln \left( \frac{\lambda^2}{\bar{c}^2} \right) - \ln \left( \frac{\lambda^2}{\bar{c}^2} + 1 \right) , \quad (120)$$

which allows us to rewrite the oTBA equations in the form

$$(a_0 + a_2) * \ln \eta_n = \tilde{d}_n + a_1 * [\ln(1 + \eta_{n-1}) + \ln(1 + \eta_{n+1})] . \quad (121)$$

We note that $\tilde{d}_n$ is in fact independent of $n$. Carrying out the Fourier transform and using that $f_0 - f_2 = (a_0 - a_2) * f_0$ we obtain

$$\begin{aligned}
\mathscr{F}[\ln \eta_n] &= \frac{1}{1 + e^{-\bar{c}|k|}} (1 - e^{-\bar{c}|k|}) \mathscr{F}[f_0] \\
&+ \frac{1}{1 + e^{-\bar{c}|k|}} e^{-\frac{\bar{c}|k|}{2}} \mathscr{F}[(\ln(1 + \eta_{n-1}) + \ln(1 + \eta_{n+1}))] .
\end{aligned} \quad (122)$$

The first term on the right hand side simplifies

$$\frac{1}{1 + e^{-\bar{c}|k|}} (1 - e^{-\bar{c}|k|}) \mathscr{F}[f_0] = -2\pi \frac{\tanh(\bar{c}k/2)}{k} . \quad (123)$$

Finally, taking the inverse Fourier transform of (122), using (110) as well as

$$\int_{-\infty}^{\infty} dk \, e^{-ik\lambda} \frac{\tanh(\bar{c}k/2)}{k} = -\ln \left[ \tanh^2 \left( \frac{\pi \lambda}{2\bar{c}} \right) \right] , \quad (124)$$

we arrive at the desired tri-diagonal form of the oTBA equations

$$\ln(\eta_n) = d + s * [\ln(1 + \eta_{n-1}) + \ln(1 + \eta_{n+1})] , \qquad n \geq 1 , \tag{125}$$

$$\eta_0(\lambda) = 0 . \tag{126}$$

Here $s(\lambda)$ is given by Eq. (112) and

$$d(\lambda) = \ln \left[ \tanh^2 \left( \frac{\pi \lambda}{2c} \right) \right]. \tag{127}$$

## C  Asymptotic behaviour

In this appendix we derive the asymptotic condition (47) for the tri-diagonal equations (44). Our derivation closely follows the finite temperature case [75]. We start from Eq. (42) for $n = 1$

$$\ln \eta_1(\lambda) = -2h + (f_0 + f_1) + a_2 * \ln(1 + \eta_1^{-1}) + \sum_{m=2}^{+\infty} (a_{m-1} + a_{m+1}) * \ln\left(1 + \eta_m^{-1}\right) , \tag{128}$$

where $f_r = f_r(\lambda)$ is defined in (119). We use now the following identities, which are easily derived from (118), (120), (121)

$$\begin{aligned} a_2 * \ln(1 + \eta_1^{-1}) &= a_2 * \ln(1 + \eta_1) - a_2 * \ln \eta_1 = \\ &= a_2 * \ln(1 + \eta_1) - f_0 + f_2 - a_1 * \ln(1 + \eta_2) + \ln \eta_1 . \end{aligned} \tag{129}$$

Using (129) we can recast (128) in the form

$$\begin{aligned} -2h + a_1 * (f_0 + f_1) = a_1 * \ln \eta_2 \quad &- \quad a_2 * \ln(1 + \eta_1) - a_3 * \ln(1 + \eta_2^{-1}) \\ &- \sum_{m=3}^{+\infty} (a_{m-1} + a_{m+1}) * \ln(1 + \eta_m^{-1}) . \end{aligned} \tag{130}$$

To proceed, we write

$$\begin{aligned} \sum_{m=3}^{+\infty} (a_{m-1} + a_{m+1}) * \ln(1 + \eta_m^{-1}) &= (a_2 + a_4) * \ln(1 + \eta_3^{-1}) \\ &+ \sum_{m=4}^{+\infty} (a_{m-1} + a_{m+1}) * \ln(1 + \eta_m^{-1}) . \end{aligned} \tag{131}$$

After rewriting the first term on the right hand side, we substitute back into (130) to obtain

$$\begin{aligned} -2h + a_2 * (f_0 + f_1) = a_2 * \ln \eta_3 \quad &- \quad a_3 * \ln(1 + \eta_2) - a_4 * \ln(1 + \eta_3^{-1}) \\ &- \sum_{m=4}^{+\infty} (a_{m-1} + a_{m+1}) * \ln(1 + \eta_m^{-1}). \end{aligned} \tag{132}$$

Iterating the above procedure $n$ times we arrive at

$$\begin{aligned} -2h + a_n * (f_0 + f_1) = a_n * \ln \eta_{n+1} \quad &- \quad a_{n+1} * \ln(1 + \eta_n) - a_{n+2} * \ln(1 + \eta_{n+1}^{-1}) \\ &- \sum_{m=n+2}^{+\infty} (a_{m-1} + a_{m+1}) * \ln(1 + \eta_m^{-1}) . \end{aligned} \tag{133}$$

Fourier transforming and using the definition for $f_r$ given in (119) we obtain

$$
\begin{aligned}
\ln \eta_{n+1} =\ & -2h + \ln\left[\frac{\lambda}{\bar{c}}\left(\frac{\lambda^2}{\bar{c}^2} + \frac{1}{4}\right)\right] + a_1 * \ln \eta_n \\
& + a_1 * \ln(1 + \eta_n^{-1}) + a_2 * \ln(1 + \eta_{n+1}^{-1}) + \sum_{m=2}^{+\infty}(a_{m-1} + a_{m+1}) * \ln(1 + \eta_{m+n}^{-1}). \quad (134)
\end{aligned}
$$

Assuming that $\eta_n^{-1}(\lambda)$ is vanishing sufficiently fast as $n \to \infty$ for a generic (and fixed) value of $\lambda$, we can drop the infinite sum and the two previous terms, and arrive at Eq. (47).

## D Perturbative analysis

In this appendix we sketch the calculations leading to the expansion (63). Throughout this appendix we work with the dimensionless variable $x = \lambda/\bar{c}$. At the lowest order, it follows from Eq. (62) that

$$
\varphi_1(x) = \frac{\tau^2}{x^2\left(x^2 + \frac{1}{4}\right)} + \mathcal{O}(\tau^3). \quad (135)
$$

Since $\varphi_n(x) \propto \tau^{2n}$, we can neglect $\varphi_n(x)$ with $n \geq 2$ to compute the third order expansion of $\varphi_1(x)$. Hence, the infinite sum in (59) for $n = 1$ can be truncated, at third order in $\tau$, to the first term ($m = 1$), where we can use the expansion (135) for $\varphi_1(\lambda)$. Following Ref. [51] one can then use identity (118) to perform the convolution integral and finally obtain

$$
\varphi_1(x) = \frac{\tau^2}{x^2\left(x^2 + \frac{1}{4}\right)}\left(1 - \frac{4\tau}{x^2 + 1}\right) + \mathcal{O}(\tau^4). \quad (136)
$$

One can then perform the same steps for higher order corrections, at each stage of the calculation keeping all the relevant terms. For example, already at the fourth order in $\tau$ of $\varphi_1(x)$ one cannot neglect the lowest order contribution coming from $\varphi_2(x)$ in the r.h.s. of Eq. (59). For higher orders one also has to consider corrections to $\varphi_n(x)$ with $n \geq 2$.

## E Small $\gamma$ limit for $g_2$

In this appendix we prove that

$$
\lim_{\gamma \to 0} g_2(\gamma) = 2 . \quad (137)
$$

Our starting point is Eqn (86). Rescaling variables by

$$
\hat{b}_m(x) = \sqrt{\gamma}\, b_m\left(\frac{x}{\sqrt{\gamma}}\right), \qquad \hat{\eta}_n(x) = \tilde{\eta}_n\left(\frac{x}{\sqrt{\gamma}}\right), \quad (138)
$$

we have

$$
g_2 = 2 + \sqrt{\gamma} \sum_{m=1}^{\infty} \int_{-\infty}^{\infty} \frac{dx}{2\pi}\left[2mx\, \hat{b}_m(x)\frac{1}{1 + \hat{\eta}_m(x)}\right]. \quad (139)
$$

The functions $\hat{b}_n(x)$ satisfy the coupled nonlinear integral equations

$$
\hat{b}_n(x) = nx - \sum_{m=1}^{\infty} \int_{-\infty}^{\infty} dy\, \frac{1}{1 + \hat{\eta}_m(y)}\hat{b}_m(y)\hat{a}_{nm}(x - y) , \quad (140)
$$

where

$$\hat{a}_{nm}(x) = \frac{1}{\sqrt{\gamma}} \widetilde{a}_{nm}\left(\frac{x}{\gamma}\right). \tag{141}$$

Our goal is to determine the limit

$$\lim_{\gamma \to 0} \sum_{m=1}^{\infty} \int_{-\infty}^{\infty} \frac{\mathrm{d}x}{2\pi} \left[ 2mx\,\hat{b}_m(x) \frac{1}{1+\hat{\eta}_m(x)} \right]. \tag{142}$$

The calculation is non-trivial as we cannot exchange the infinite sum with the limit. However, based on numerical evidence we claim that this limit is finite, and (137) then immediately follows from (139).

Note that the numerical computation of $g_2(\gamma)$ is increasingly demanding as $\gamma \to 0$, due to the fact that more and more strings contribute. Accordingly, the infinite systems (83) and (84) have to be truncated to a larger number of equations and the numerical computation takes more time to provide precise results. We were able to numerically compute $g_2(\gamma)$ for decreasing values of $\gamma$ down to $\gamma = 0.025$ where $g_2(0.025) \simeq 2.11$ and approximately 30 strings contributed to the computation. We fitted the numerical data for small $\gamma$ with $G(\gamma) = \alpha_1 + \alpha_2 \sqrt{\gamma}$ and we correctly found $\alpha_1 = 2$ within the numerical error.

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
