# Peer review of "Quantum quenches to the attractive one-dimensional Bose gas: exact results"

_SciPost Physics, doi:SciPost Phys. 1, 001 (2016)_

## Round 2 · Referee Report · Anonymous · 2016-6-8

Strengths

1- Exact results.
2- Experimentally relevant.
3- Well written, clear exposition.
4- Exceptionally good (wide) list of references.

Weaknesses

I don't see any weakness. It is true that most of the theoretical methods are already existing, but they are applied to a new situation, and there are certain steps that required new ideas.

Report

This paper derives exact solutions for a specific quench problem in the attractive Lieb-Liniger model. The methods are simple generalizations of already available techniques, however, there are certain innovations and all results are new. The g2 local pair correlator is calculated numerically, and it is studied in the small coupling and large coupling limits. The physical consequences of the bound states are discussed. Maybe the most interesting is that g2 does not vanish in the infinite coupling limit. Whereas this is not necessarily surprising, it is useful that exact results are derived. I believe that later the results could be compared to experiments, thus strengthening the link between theory and experiment.

Requested changes

I don't request any changes, but I have a comment to the authors. It would be certainly interesting to consider the higher local correlators as well. For example g3. Maybe the three-strings would show up there in the large coupling limit?

  • validity: top
  • significance: high
  • originality: good
  • clarity: top
  • formatting: perfect
  • grammar: perfect

Author:  Lorenzo Piroli  on 2016-07-06  [id 48]

(in reply to Report 1 on 2016-06-08)

The referee states that

“It would be certainly interesting to consider the higher local correlators as well. For example g3. Maybe the three-strings would show up there in the large coupling limit?”

We agree that the calculation of g3 would be very interesting, and three-strings may well give non-vanishing contributions in the large coupling limit. 
While g3 is known for general states in the repulsive Lieb-Liniger model, the attractive case is significantly harder to treat due to the presence of bound states and requires further development of existing methods. We think that such a calculation is possible, but definitely beyond the scope of our paper.

---

## Round 2 · Referee Report · Anonymous · 2016-6-27

Strengths

1. Extremely well written and free of typos.

2. Generously referenced.

3. Technical details of the computations presented in the manuscript are given
in full.

Weaknesses

1. While the computations in the manuscript are technically impressive, the discussion of the physical consequences of these computations is not as expansive as it might be.

Report

This manuscript studies the long time behaviour after a quantum quench in the Lieb-Liniger model, a model of a one-dimensional Bose gas. The particular quench studied takes the gas from c, the strength of its two-body interaction, zero to c finite and negative (i.e. the gas is quenched into its attractive regime).

The authors use the quench action to study this quench. The needed overlaps between the eigenstates of the post-quench Hamiltonian and the initial state of the gas had been determined by other authors. However some additional analysis of the overlaps in the case of states that are zero momentum n-strings by the authors was required. With the overlaps in hand, the authors are able to analyze the corresponding generalized TBA equations describing the steady state post-quench. Impressively, the authors are able to arrive at an analytical solution to these equations. Finally the authors use this solution to investigate g2, the pair correlation function (which can be obtained as a derivative of the generalized free energy by c). Interestingly the authors show that the density of higher bound states is reduced by increasing the strength of the attractive interaction.

I appreciate the careful presentation of results in this paper. It reads nicely and it is good to see
that all the details of the computations are presented in a comprehensible fashion. The one critique
that I do have is the sole focus on the steady state. It would be interested to understand more about
the approach to steady state. There have been suggestions in the literature (indeed, involving
one of the authors) of unusual entanglement entropy growth after a quench in systems with bound states.
It would be interesting to know if the authors felt something like this would occur here as well. I am not necessarily asking for a detailed computation. But some remarks to this end would be appreciated.

Requested changes

1. As I indicated, I would like to see some comments and/or analysis on the approach of the gas to the steady state post-quench.

  • validity: top
  • significance: good
  • originality: ok
  • clarity: top
  • formatting: excellent
  • grammar: excellent

Author:  Lorenzo Piroli  on 2016-07-06  [id 49]

(in reply to Report 2 on 2016-06-27)

To understand the approach to the stationary state after quantum quenches is without doubt a very interesting and timely problem.
However, it is well known that this is an extraordinarily difficult task to study with integrability methods in truly interacting models. Even the case of the Transverse Field Ising chain, which can be mapped to a free fermion theory, is extremely non-trivial as is clear from the works of Calabrese, Essler and Fagotti.
In the interacting case there are basically only two results available in the literature: one is for particular quenches in the sine-Gordon model (http://arxiv.org/pdf/1405.4813.pdf, one of us is among the authors), and the other one is a semi-numerical paper ( http://arxiv.org/abs/1505.03080, one of us is among the authors ) studying the approach to the stationary state for particular observables in the Lieb-Linger model in the regime where no string solutions exist. The case of interest to us here is more difficult than either of these two, and studying the time evolution is manifestly beyond the scope of our work.

The referee also mention the anomalous behavior of entanglement entropy and correlation functions found in http://arxiv.org/abs/1604.03571 (of which another one of us is an author). This novel effect is due to the phenomenon of confinement, i.e. not to the presence of the bound states, but  to the absence of elementary excitations. In our case, unbound particles are always present as $\rho_1$ is always different from zero. This means that in the quench studied here this phenomenon does not take place.

For the reasons given above we decided not to change the manuscript by adding comments on the finite time dynamics.

---

## Editorial Decision

published